# Screening of Serum Biomarkers of Coal Workers’ Pneumoconiosis by Metabolomics Combined with Machine Learning Strategy

**DOI:** 10.3390/ijerph19127051

**Published:** 2022-06-09

**Authors:** Zhangjian Chen, Jiaqi Shi, Yi Zhang, Jiahe Zhang, Shuqiang Li, Li Guan, Guang Jia

**Affiliations:** 1Department of Occupational and Environmental Health Sciences, School of Public Health, Peking University, Beijing 100191, China; zhangjianchen@pku.edu.cn (Z.C.); 1610306221@pku.edu.cn (J.S.); 1710306142@pku.edu.cn (Y.Z.); zhangjh_pku15@126.com (J.Z.); 2Department of Occupational Disease, Peking University Third Hospital, Beijing 100191, China; shuqiangli@263.net

**Keywords:** pneumoconiosis, metabolomics, biomarkers, case–control study, machine learning

## Abstract

Pneumoconiosis remains one of the most serious global occupational diseases. However, effective treatments are lacking, and early detection is crucial for disease prevention. This study aimed to explore serum biomarkers of occupational coal workers’ pneumoconiosis (CWP) by high-throughput metabolomics, combining with machine learning strategy for precision screening. A case–control study was conducted in Beijing, China, involving 150 pneumoconiosis patients with different stages and 120 healthy controls. Metabolomics found a total of 68 differential metabolites between the CWP group and the control group. Then, potential biomarkers of CWP were screened from these differential metabolites by three machine learning methods. The four most important differential metabolites were identified as benzamide, terazosin, propylparaben and N-methyl-2-pyrrolidone. However, after adjusting for the influence of confounding factors, including age, smoking, drinking and chronic diseases, only one metabolite, propylparaben, was significantly correlated with CWP. The more severe CWP was, the higher the content of propylparaben in serum. Moreover, the receiver operating characteristic curve (ROC) of propylparaben showed good sensitivity and specificity as a biomarker of CWP. Therefore, it was demonstrated that the serum metabolite profiles in CWP patients changed significantly and that the serum metabolites represented by propylparaben were good biomarkers of CWP.

## 1. Introduction

Pneumoconiosis is one of the most serious global occupational diseases and includes a group of respiratory diseases caused by the inhalation of mineral dust. The Global Burden of Disease Study (GBD) demonstrated that the global incidence of pneumoconiosis increased by 66.0% from 1990 to 2017 [1]. As coal still plays a dominant part in global energy production and consumption, there is a very large number of people exposed to coal dust [2]. More than 50% of annual officially reported occupational cases were coal workers’ pneumoconiosis (CWP) in China, which may still be underestimated due to insufficient occupational health examination and strict diagnostic criteria [3]. At present, the diagnosis of pneumoconiosis mainly refers to a reliable occupational exposure history of dust and chest X-ray radiographs, lacking some objective and early stage biomarkers. Meanwhile, as it generally takes 10–20 years to develop pneumoconiosis after exposure to dust and pneumoconiosis is difficult to cure once you suffer from it, exploring biomarkers for early diagnosis is also very essential for better prevention of CWP.

The etiologies of CWP have been confirmed, which is a pulmonary fibrosis disease caused by exposure to various coal dusts [4,5,6]. The pattern and disease progression of CWP are directly related to the physicochemical properties, types and concentrations of exposed coal dust or silica [7,8,9,10]. However, the complex dust etiology and long latent period make it difficult to detect CWP early. At present, there is also a lack of sensitive and specific biomarkers, so that it is difficult to dynamically observe the disease evolution of CWP. Metabolomics, as a system biology method simultaneously detecting thousands of metabolites with high throughput, focuses on the metabolic changes in biological systems [11]. With the development of omics technology, metabolomics has been widely used for biomarker discovery and mechanism exploration [12]. Meanwhile, metabolomics is downstream of multiomics and is considered to be an extension of genomics and proteomics, making the amplified signal of metabolite easier to detect than gene and protein [13]. The applications of machine learning have been widely carried out in biomedicine toward improved diagnosis and treatment [14]. Machine learning combined with metabolomics can more precisely screen potential biomarkers for disease such as early stage lung adenocarcinoma from high-throughput data [15].

The present study aimed to identify potential sensitive serum biomarkers associated with occupational coal workers’ pneumoconiosis (CWP) by a case–control study using metabolomics combined with a machine learning (ML) strategy. Untargeted metabolomics was conducted in high-performance liquid chromatography–mass spectrometry (HPLC–MS) to analyze the changes in serum metabolite profiles of occupational pneumoconiosis patients compared with healthy controls. The potential biomarkers of CWP were screened from differential metabolites by a series of machine learning methods.

## 2. Materials and Methods

### 2.1. Study Design and Subjects

This case–control study included 150 coal workers’ pneumoconiosis (CWP) patients with different stages from two representative occupational disease specialist hospitals in Beijing, China from January to December 2021. In addition, 120 healthy controls from an authoritative health examination institution in Beijing were also included. All study subjects were involved according to the strict inclusion and exclusion criteria. The inclusion criteria for the case group were clinically diagnosed CWP patients with clear occupational dust exposure history covering stage 1, stage 2 and stage 3 that represents the severity of CWP. The staging of CWP was mainly distinguished by the appearance of X-ray in the national pneumoconiosis diagnostic criteria. The subjects in the control group were all healthy people with no occupational dust exposure history in Beijing, who matched the case group with age, smoking, drinking, gender and place of residence as much as possible. The exclusion criteria for the case and control group were the exclusion of subjects with lung cancer and various other diffuse pulmonary fibrosis diseases; exclusion of subjects with recent history of major respiratory diseases, including chest trauma or surgery affecting the lungs in the past 1 year, pneumonia, pleurisy, emphysema anda asthma; and limiting subjects with nonrespiratory chronic diseases. Information on demographic characteristics such as age, gender, smoking and drinking, as well as occupational history and working age were investigated by a detailed questionnaire. The present study was approved by the Peking University Third Hospital Medical Science Research Ethics Committee (Approval number: M2024504), and informed consent was obtained from each subject.

### 2.2. Serum Sample Collection

Serum samples were collected from all subjects by drawing morning fasting blood from the cephalic veins in the arm. Each subject was given 2 mL of venous blood. Then, blood samples were allowed to stand at room temperature for 3 h and centrifuged for 7 min (3000 rpm) to obtain serum. Before use, the final serum sample was placed in a cryotube and immediately stored in a refrigerator at −80 °C.

### 2.3. Serum Metabolomics

Sample preparation. At first, 100 µL serum was added to 400 μL precooled chloroform/methanol (2:1) solution. After centrifugation (14,000× *g*, 4 °C) for 15 min, the supernatant was transferred to a new centrifuge tube and vacuum freeze-dried. Each sample was redissolved in 50 μL acetonitrile (50%) before testing on the machine. Finally, 10 μL of each sample was taken for sample detection and 5 μL of each sample was taken for quality control (Qc) samples. 

HPLC–MS analysis for untargeted metabolomics. A high-performance liquid chromatography–mass spectrometry (HPLC–MS) system (U3000, Thermo, Waltham, MA, USA) was used for untargeted metabolomics analysis. The parameters of the chromatographic column (Xbridge amide column, Waters, Milford, MA, USA) were as follows: the temperature was 30 °C; the flow rate was 0.5 mL/min; mobile phase A was 5 mmol/L ammonium acetate, 95% water, 5% acetonitrile; mobile phase B was acetonitrile. The elution gradient was as follows: 0 min, 90% B; 3 min, 30% B; 12 min, 2% B; 15 min, 2% B; 16 min, 90% B; and 23 min, 90% B. An electrospray ionization (ESI) carried out the ionization of mass spectrometry. Positive and negative ion modes were implemented. Meanwhile, both primary and secondary mass spectra were collected. The primary mass spectrometer collects all metabolite information in the range of 50–750 m/z (resolution, 30,000), and then the 10 strongest peaks in the primary mass spectrum were selected for secondary mass spectrum acquisition. The dynamic collision energies were 15, 30 and 45, and the resolution was 15,000.

Annotation of mass spectrometry data. After raw data preprocessing by MS-DIAL software, annotation of the MS data was conducted by the MassBank database. The error ranges of primary mass spectrometry and secondary mass spectrometry were set as 0.01 Da and 0.05 Da, respectively. The cutoff score for final identification was 70%. The metabolite codes were searched in the human metabolomics database (HMDB).

Data analysis of metabolomics. Multivariate analysis of metabolomics data was carried out by SIMCA15.0.2 software. The principal component analysis (PCA) as a supervised model was conducted to show the different trends of metabolomics characteristics between the CWP case group and control group. The orthogonal partial least squares discriminant analysis (OPLS-DA) as an unsupervised model was performed at the same time, along with permutation tests to verify the stability. Univariate statistics including student’s t test or Mann–Whitney U test was also performed to analyze the differential metabolites between the CWP case group and control group. The differential metabolites were shown by volcano plots, regarding false discovery rate (FDR) *p* < 0.05, absolute value of log_2_ fold change (FC) > 0.25 and the variable importance in the projection (VIP) value > 1 as the criteria. Meanwhile, metabolic pathway analysis consisted of pathway enrichment analysis and pathway topology analysis. Through the differential metabolites from metabolomics data of CWP cases and pathway data in Homo sapiens (Human) pathway libraries (hsa), significantly affected metabolic pathways were distinguished by Metaboanalyst 4.0, regarding *p* < 0.05 and pathway impact > 0.20 as the criteria.

### 2.4. Biomarker Screening by Machine Learning Strategy

Potential biomarkers of CWP were further screened from the differential metabolites analyzed by metabolomics. Machine learning (ML) strategy was used to rank importance in order to reduce dimension. Three ML methods, including random forest (RF), support vector machines (SVM) and boruta, were combined. The theoretical basis of random forest (RF) is to combine a series of weak learners (multiple decision trees) through integrated learning, so as to obtain a strong learner with significantly improved performance. The importance of features and their ranking can be obtained by RF. In this study, the mean decrease Gini index was used within RF to quantify, rank and screen the importance of each metabolite (the top three important metabolites). The greater the index of metabolites, the more important the metabolites are for the model to distinguish different groups. Support vector machine (SVM) can reduce over fitting risk by properly selecting kernel function and regularization. The linear kernel was used as the default kernel function. SVM recursive feature elimination algorithm was used in this study, referring to the previous literature [16]. The general process of the algorithm was as follows: (1) A SVM model was fitted using the specified differential metabolites as independent variables and grouping information as dependent variables; (2) Calculate the importance weight of each metabolite; (3) If the number of remaining metabolites is less than the specified number of metabolites, end the screening and return the importance weight of each metabolite; (4) Eliminate the metabolites with the lowest weight; (5) Repeat (1)–(4). The specified number of metabolites is set to 10 by default. Boruta is an RF based feature screening method, selecting key features that have significant discrimination ability than random displacement features. The maximum number of RF runs was 1000. When temporary features were included, secondary selection would be carried out to judge whether some metabolites with large fluctuation should be included in the selected features. The differential metabolites as potential biomarkers of CWP must be confirmed by boruta.

At last, the top three metabolites of RF and SVM results as well as the top three metabolites confirmed by boruta results were combined as potential biomarkers of CWP. Then, to adjust for the influence of potential confounding factors, including age, smoking, drinking and chronic diseases, multiple logistic regression analysis was conducted to confirm the association between the potential biomarkers and different stages of CWP. Finally, the receiver operating characteristic (ROC) curve of the final CWP biomarker was analyzed. The sensitivity and specificity were judged by the area under the curve (AUC).

### 2.5. Statistical Analysis

Data were expressed as the means ± SD or quantity (percentage). SPSS 20.0 software was used to analyze the data other than metabolomics. Statistical differences in continuous variables between two independent groups were analyzed by the Student’s t test or Mann–Whitney U test. Statistical differences in categorical variables were determined by comparing the rates using Pearson χ^2^ test. The *p* < 0.05 was considered as criteria for determining statistically significant differences.

## 3. Results

### 3.1. The Characteristics of Subjects

A total of 150 male coal workers’ pneumoconiosis (CWP) cases and 120 male controls were involved in the present study. The characteristics of the subjects in these two groups were shown in Table 1. The results showed that there were significant differences in age, smoking status and chronic diseases between the two groups (*p* < 0.05), but no difference existed in drinking status. The subjects in the CWP case group had a higher average age and higher smoking and chronic disease rates than the controls. The CWP patients all had a coal-related occupation history, with an average working experience age of 24.70 ± 8.48 years. The cases included stage 1, stage 2 and stage 3 CWP, accounting for 62.7%, 31.3% and 6.0%, respectively. Meanwhile, only 26.0% of cases had no complications, while the complications of tuberculosis, COPD and chronic bronchitis accounted for 12.7%, 21.3% and 37.3%, respectively. There were also four (2.7%) cases of two complications. Both the CWP cases and controls were male, and both were from Beijing, who lived and worked in Beijing most of the time.

### 3.2. Differential Metabolites between the CWP Case Group and Control Group

The differences in metabolomics profiles between the CWP case group and control group were analyzed, and then differential metabolites were distinguished to screen potential biomarkers. There were 345 metabolites identified by annotation of MS data in the serum of subjects. Significant overall differences in metabolomics profiles between the case and control groups were found in both PCA (Figure 1A) and OPLS-DA results (Figure 1B). In both supervised and unsupervised models, there was an obvious separation trend between cases and controls. Meanwhile, the results of permutation test demonstrated that the unsupervised OPLS-DA models were very good (Figure 1C). Then, a total of 68 differential metabolites were identified from the intersection of 105 differential metabolites from the multidimensional statistics (Figure 1D) and 117 differential metabolites from the univariate statistics (Figure 1E). The detailed information including name, class and a series of statistical indicators on these 68 identified differential metabolites (Figure 1F) was shown in Appendix A. The standard score (Z score) map showed that the relative expression of 39 differential metabolites (57.4%) increased in the CWP case group, while other 29 differential metabolites (42.6%) decreased (Appendix A). The top seven metabolites with the greatest difference (fold change (FC) > 4 and log_2_FC > 2) were propylparaben, (s,s)-(+)-tetrandrine, benzamide, N-methyl-2-pyrrolidone, aminopyrine, perfluorooctanoic acid and salicylic acid (Appendix A). Compared to the control group, the relative expression of these top seven metabolites all increased in the CWP case group.

### 3.3. Pathway Analysis of Serum Metabolomics

Pathway analysis of serum metabolomics was shown in Figure 2. Only the phenylalanine metabolism pathway was enriched, but it was not significantly impacted (pathway impact = 0) in the pathway topology analysis. The pathway map of phenylalanine metabolism from KEGG was shown in Appendix A. However, after Holm or FDR correction to reduce the false positive rate, it was found that the change in the phenylalanine metabolism pathway was not statistically significant. Therefore, although a variety of differential metabolites were found in CWP cases, it was unable to focus them on certain metabolic pathways.

### 3.4. Screening Potential Biomarkers of CWP

Potential biomarkers of CWP were further screened from the differential metabolites by machine learning (ML) strategy. Three ML methods, including RF, SVM and boruta, were used to rank importance. The top three metabolites of RF results were benzamide, tetrazosin and propylparaben (Figure 3A). The top three metabolites of SVM results were propylparaben, benzamide and N-methyl-2-pyrrolidone (Figure 3B). Then, boruta was used for further screening, and 50 differential metabolites were confirmed, among which benzamide, tetrazosin and propylparaben were the top three (Figure 3C). In summary, the three ML methods identified the four most important metabolites, propylparaben, benzamide, terazosin and N-methyl-2-pyrrolidone, as potential biomarkers for CWP. Moreover, the results of the three methods had strong consistency, which increased the reliability of the results. However, after adjusting for confounding factors including age, chronic diseases, smoking and drinking, logistic regression analysis confirmed that CWP was only significantly related to the relative expression of propylparaben.

### 3.5. Effect of CWP Stage on the Biomarker Screening

According to the above analysis strategy, three different stages of CWP (stage 1, stage 2 and stage 3) were analyzed for biomarker screening. The results showed that the metabolomics profiles of all stages of CWP were different from the control group (Figure 4A). Differential metabolites between the two groups showed crosses (Figure 4B). More importantly, it was found that the above potential biomarkers for CWP all belonged to the cross of three CWP stages. Furthermore, after adjusting for the influence of confounding factors, multiple logistic regression analysis confirmed that CWPs at different stages were only significantly related to propylparaben (Figure 4C). The relative contents of propylparaben in the control group and CWPs at different stages were shown in Figure 4D, which showed a good increasing trend with the severity of the disease. This result indicated that the increase in this biomarker may also reflect the severity of CWP.

### 3.6. Diagnostic Analysis of Potential Biomarkers for CWP

Diagnostic experiments were conducted for the key potential biomarker of CWP. As shown in Figure 5, the receiver operating characteristic (ROC) curve for propylparaben was drawn. The results showed that the area under the curve (AUC) reached 0.777 (95% CI: 0.717–0.837), indicating good sensitivity and specificity for the differential metabolite of propylparaben as a potential biomarker of CWP.

## 4. Discussion

This study focused on biomarkers of pneumoconiosis and achieved satisfactory results by using a case–control design and serum metabolomics detection. Significant changes in the metabolic profile in the serum occurred between the CWP patients and healthy controls. The contents of 68 metabolites changed significantly in the CWP case group. Through further analysis by machine learning strategy, the four most important differential metabolites, including propylparaben, benzamide, terazosin and N-methyl-2-pyrrolidone, were screened. However, after adjusting for the influence of confounding factors, this study identified propylparaben as a good biomarker of CWP. At the same time, the contents of serum propylparaben in different stages of CWP had a good progressive relationship, which increased with the severity of the disease. The ROC curve also confirmed the good sensitivity and specificity of this serum metabolite as a CWP biomarker. 

Metabolomics is a very suitable method to explore biomarkers [17]. In recent years, metabolomics has been widely used in the screening of biomarkers of diseases, including cancer [18,19], cardiovascular diseases [20], liver diseases [21], respiratory diseases [22], diabetes mellitus [23] and even mental illness [24]. This research not only screened out a good biomarker that could reflect CWP and its severity but also suggested that metabolomics combined with ML strategy may be an effective way to explore biomarkers in complex diseases such as pneumoconiosis. Pneumoconiosis currently lacks good objective biomarkers because the main basis for CWP diagnosis is still X-ray chest radiography, which has certain subjectivity. Meanwhile, the diagnosis of pneumoconiosis involves occupational disease compensation in China, making the diagnosis more sensitive and potential conflicts of interest. Therefore, good objective biomarkers are very important and practical for these complex diseases, such as pneumoconiosis, and this study provides a universal and practical method for their exploration.

Propylparaben is an organic compound, also known as propyl chemosept or propyl parasept, belonging to the class of benzene and substituted derivatives. The sources of human exposure to propylparaben are diverse, and the exposure time is long. Food, cosmetics and drugs are considered as the main exposure sources, accounting for 1.3%, 66% and 33%, respectively [25]. Liao et al. [26,27] detected various kinds of parabens with an average content of 39.3 ng/g in almost all foods in China and the United States (USA), and the average proportion of propylparaben was approximately 10%. Guo et al. [28,29] tested the content of parabens in many types of cosmetics in China and the USA. They found that all cosmetics contained parabens, among which methylparaben and propylparaben were the most abundant. Given that propylparaben is widely used in cosmetics, and women use more cosmetics, gender is indeed a potential confounding factor. However, due to the particularity of coal workers’ pneumoconiosis (CWP) patients, the cases are basically male. Both cases and controls included in this study were male. Cosmetics are used less in men, so we did not collect the information of cosmetic use. In addition, propylparaben also appeared in drugs [30], water [31] and indoor dust [32]. Extensive environmental exposure to propylparaben has resulted in the general situation of human intake and made it a common exogenous metabolite in the human body [33].

Propylparaben in the human body can be metabolized in the intestine and liver, partially keep the prototype, and finally be excreted from the body through urine, bile and feces. The known metabolic pathways of propylparaben may include hydrolysis, transesterification and hydroxylation [34,35]. However, many previous studies have reported that complete parabens can be detected in human serum, urine, placenta, breast milk and breast tumor tissue [36,37,38,39,40]. Ye et al. found that the median values of propylparaben in human serum and urine could reach approximately 10 μg/L, which indicated that propylparaben could keep the prototype after being absorbed, avoiding the degradation of skin vinegar enzyme and intestinal or liver metabolic systems [36,39]. The present study found that the prototype propylparaben showed different levels in the serum of CWP patients at different stages, which may be related to changes in its metabolism. However, research on the relationship between the metabolism of propylparaben and CWP development has not been reported. Pneumoconiosis is closely related to the oxidative stress of the body, and redox imbalance may interfere with the metabolism of propylparaben [41,42]. Therefore, we speculated that CWP-related redox imbalance may be one of the reasons for the metabolic disorder of propylparaben. However, the specific mechanism is still unclear and needs further study.

Currently, clinical identification and diagnosis of pneumoconiosis mainly rely on radiological images, but they are more or less subjective and late. Biomarkers of pneumoconiosis for early detection or diagnosis are of great significance for early identification and intervention of diseases. Although there have been many studies on the biomarkers of pneumoconiosis, the biomarkers with good sensitivity and specificity are still very limited, especially those verified by population studies. Many potential biomarkers are proposed based on toxic effects and underlying mechanisms following exposure to coal dust or silica, such as glutathione, glutathione peroxidase activity, TNF-α and IL-8, etc. [43]. However, these oxidative, inflammatory and immunological biomarkers are usually not very specific. Further studies also found serum Clara cell protein-16 (CC16) and l-selectin levels could be good biomarkers of pneumoconiosis effect, and neopterin levels in urine and serum may be good exposure biomarkers [44]. However, large sample epidemiological studies and are still needed for population verification, especially in combination with a large number of different types of people, including pneumoconiosis, healthy control and even nonpneumoconiosis workers exposed to occupational dust. In this study, a case–control study design and advanced metabolomics combined with machine learning methods were used to screen biomarkers of CWP, which had certain initiative and innovation. A recent multiomics study [45] that was integrated with transcriptomics and metabolomics analyses of silicosis was reported, which revealed that several metabolites were significantly upregulated in silicosis mouse lungs, such as prostaglandin D2 (PGD_2_) and thromboxane A2 (TXA_2_). The authors of this article believed that arachidonic acid (AA) pathways, especially PGD_2_ and TXA_2_, may be potential therapeutic targets for silicosis. However, the biomarkers proposed in this study still need to be tested for validity and feasibility in further studies.

The potential biomarker of propylparaben for CWP was screened by metabolomics and a series of bioinformatics analyses. Practices have indicated that these methods have great advantages and applicability. Metabolomics has been considered an excellent high-throughput method to explore biomarkers and even explore mechanisms [12,17,46]. Subtle metabolomic changes could appear before significant functional cellular damage [47], which confirmed the good sensitivity of metabolomics. However, the problem with a sufficiently sensitive method is the dimensionality reduction screening of the most important biomarkers. In the present study, ML strategy was used for precise screening. Firstly, 68 differential metabolites were analyzed by univariate and multivariate statistics from hundreds of identified metabolites in serum metabolomics. Secondly, ML methods were further used to screen more important differential metabolites. Three ML methods, including RF, SVM and boruta, have their own advantages and disadvantages [16]. RF is simple, easy to implement, fast and shows strong performance in many real tasks. However, the RF model is not easy to explain, and it may fall into “over fitting” when the sample size is small and there are many characteristics. The performance of SVM is not as good as that of neural network model and random forest. In addition, SVM is difficult to solve multi classification problems. When sample size in the metabolomics is small and differential metabolites are many, the complex kernel function may increase over fitting risk. Therefore, three ML methods were used together to identify the most important potential biomarkers. As the influence of confounding factors must be considered in epidemiological research, logistic regression analysis was performed to confirm the relationship between CWP and the relative expression of potential biomarkers after adjusting for the influence of age, chronic diseases, smoking and drinking. In this way, only one potential biomarker was retained, which was propylparaben. Finally, the sensitivity and specificity of propylparaben as a CWP biomarker were evaluated by drawing the receiver operating characteristic curve (ROC), and it was proven to have good performance. This series of methods has good effectiveness through the practice of this study and may play a role in the study of biomarkers of other diseases.

The potential public health implications of this study lie in providing biomarkers for early screening of CWP. This has important practical significance for pneumoconiosis, which has a very long incubation period and is still incurable. In addition, the metabolite biomarkers obtained in this study can also be used to complement potential laboratory diagnostic indicators of CWP, which may have better objectivity than X-ray observations. At the same time, the series of detection and data analysis methods used in this study can also provide a paradigm for the biomarker screening of other complex environment-related diseases. However, some limitations existed in the present study. The first limitation was the mismatch between the case group and the control group. Although the inclusion of the control group and the case group tried to match age, chronic diseases, living environment, smoking, drinking and other factors in the initial stage of the study design, a complete match could not be achieved in the research process. The case and control were chosen from the same area in Beijing, so we thought that the influence of the living environment was basically controlled. However, patients with CWP were older, and smoking or suffering from chronic diseases were more common than the control, so these aspects could not be completely matched in the two groups. Therefore, in the end, we must adopt statistical methods (multiple logistic regression analysis) to control their influence as much as possible. Secondly, the sample size may be also a limitation. The current sample size may be acceptable for metabolomics research and we did find a significant difference between the case group and the control group under this sample size. However, to increase the strength of the evidence for the conclusions of this study, a larger sample size study is still needed. Subsequent verification in more CWP patients would be of great significance. Finally, the complications of CWP patients and the more common chronic diseases due to older age may lead to the fact that the biomarkers screened in this study may not specifically reflect the effects of CWP. However, we believe that at least the biomarkers should be the result of the combined effect of CWP and other factors, as it is a common feature of pneumoconiosis patients. Moreover, the difference between metabolic perturbations, mainly referring to the differential expression of metabolites in metabolomics, and clinically significant changes in diseases cannot be ignored. 

## 5. Conclusions

In conclusion, the present study found that the serum metabolomics profile of CWP patients changed significantly. A total of 68 differential metabolites were identified in the serum of CWP cases compared to the control group. A series of statistical analyses, including machine learning methods, demonstrated that the serum metabolite propylparaben should be an excellent potential biomarker of CWP. At the same time, the propylparaben content was also significantly positively correlated with the stage of CWP, which could reflect the severity of the disease. However, the biological basis of this potential biomarker is still unclear. Metabolomics combined with a machine learning strategy should be an effective tool to explore biomarkers for complex diseases, including pneumoconiosis.

## Figures and Tables

**Figure 1 ijerph-19-07051-f001:**
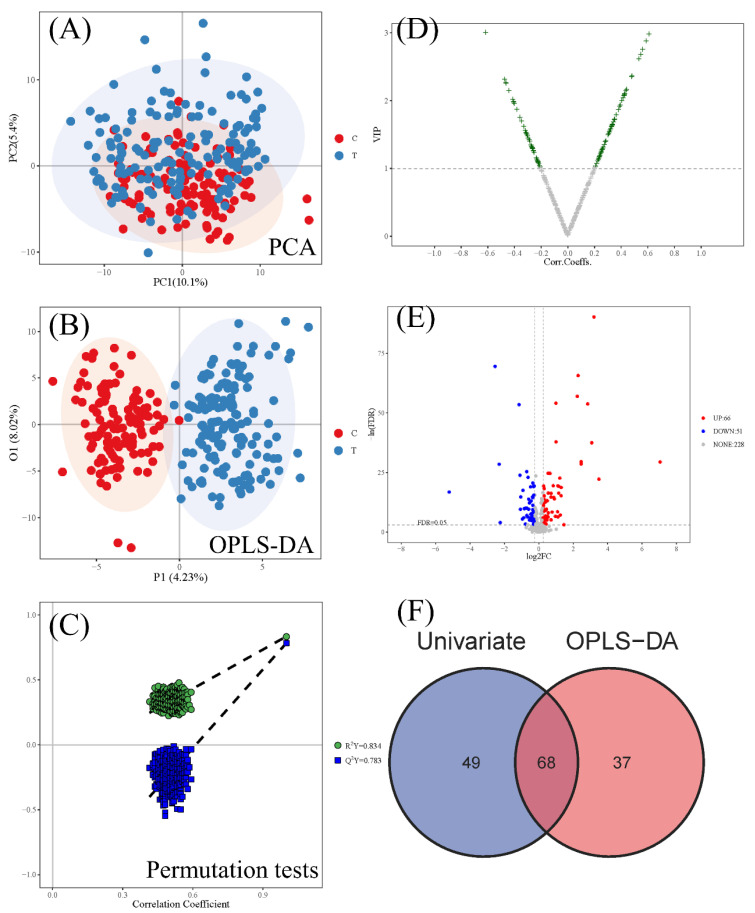
Multidimensional analysis of metabolomics profiles between the coal workers’ pneumoconiosis (CWP) case group and control group. Score scatter plot of the PCA model (**A**) and OPLS-DA model (**B**) for total metabolites. The red point: control group (**C**); the blue point: CWP case group (T). Significant overall differences in metabolomics profiles were found in both PCA and OPLS-DA results. The permutation test demonstrated that the unsupervised OPLS-DA models were very good (**C**). Differential metabolite selection between the control group and CWP case group from multidimensional statistics (**D**) and univariate statistics (**E**). For multidimensional statistics, the selection criterion of differential metabolites was a VIP (variable important in projection) value > 1. For univariate statistics, the selection criteria were FDR *p* < 0.05 and log_2_FC > 0.25. Finally, the intersection of differential metabolites from the multidimensional statistics (OPLS-DA) and univariate statistics were taken, as shown in the Venn diagram (**F**).

**Figure 2 ijerph-19-07051-f002:**
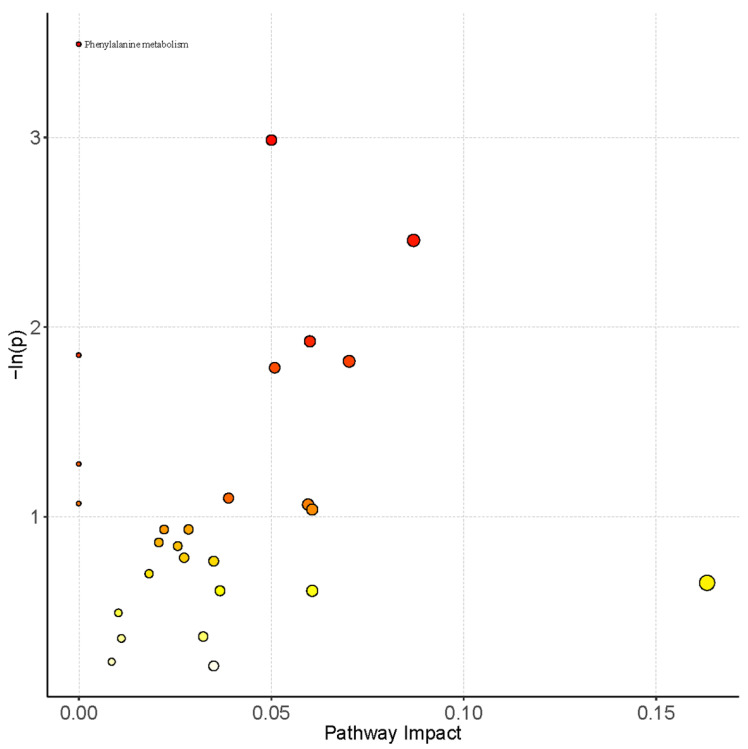
Pathway analysis of serum metabolomics between the CWP case and control groups. CWP-related metabolic pathways were shown in the pathway bubble plot, which was mapped by the combination of pathway enrichment analysis (*p* value of *Y*-axis) and pathway topology analysis (pathway impact of *X*-axis). Only the metabolic pathway of phenylalanine metabolism was enriched, but its pathway impact equaled 0.

**Figure 3 ijerph-19-07051-f003:**
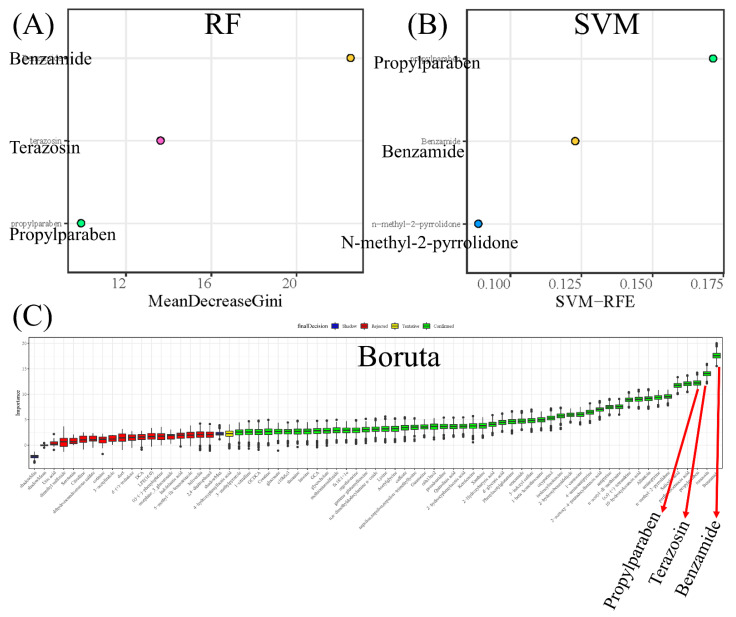
Screening potential biomarkers for coal workers’ pneumoconiosis (CWP) using machine learning (ML) strategy from differential metabolites in serum. Three ML methods were analyzed simultaneously for differential metabolites. Then, the top three metabolites of RF (**A**), SVM (**B)** and boruta (**C**) results were combined. Finally, four potential biomarkers were screened out, including propylparaben, benzamide, tetrazosin and N-methyl-2-pyrrolidone. RF: random forest; SVM: support vector machines.

**Figure 4 ijerph-19-07051-f004:**
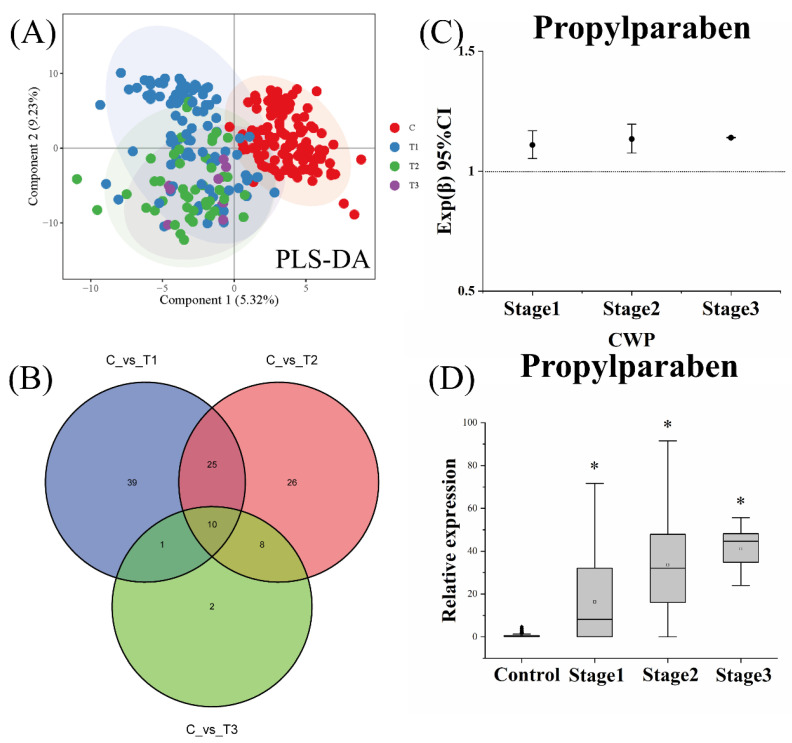
Metabolomics analysis of different stages of coal workers’ pneumoconiosis (CWP). Comparison of metabolomics profiles and analysis of differential metabolites between the different stages of CWP case groups and control group. PLS-DA (**A**) plots showed the different metabolomics profiles between different groups. The distribution of differential metabolites in different groups was also shown in the Venn diagram (**B**). T1: stage 1 CWP case group; T2: stage 2 CWP case group; T3: stage 3 CWP case group; C: control group. (**C**) After adjusting for the influence of confounding factors, including age, smoking, drinking, chronic diseases and other potential biomarkers, by using multiple logistic regression analysis, only the relative content of propylparaben was significantly related to the CWP at different stages. (**D**) The more serious the CWP disease was, the higher the content of propylparaben was. * *p* < 0.05, significant difference compared with the control group.

**Figure 5 ijerph-19-07051-f005:**
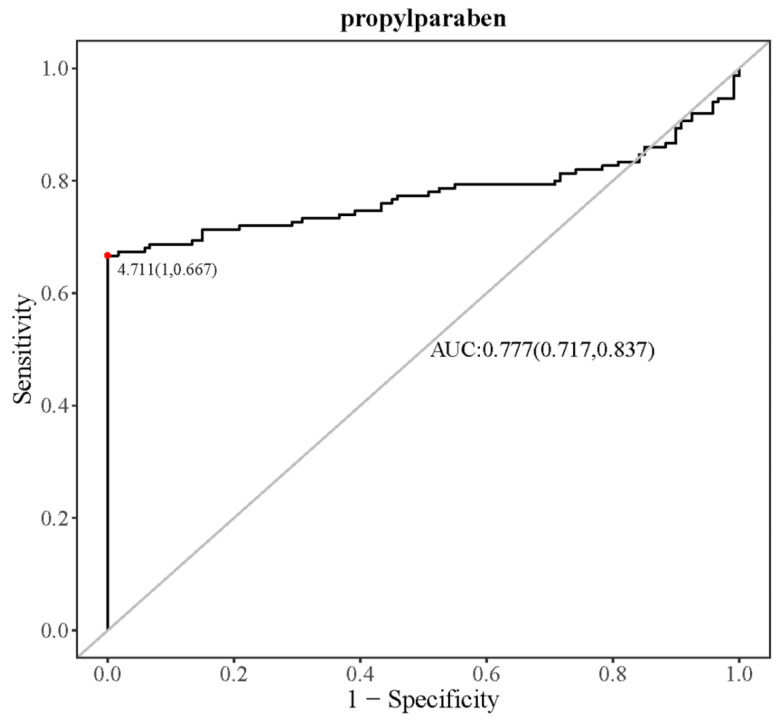
The ROC curve of propylparaben as a biomarker of CWP. The AUC indicated good sensitivity and specificity.

**Table 1 ijerph-19-07051-t001:** The descriptive analysis of characteristics of subjects in the coal workers’ pneumoconiosis (CWP) case group and control group.

	Control Group (n = 120)	CWP Case Group (n = 150)	*p*
Age (years)	56.63 ± 3.03	69.02 ± 9.07	<0.001 *
Gender			
Male	120 (100%)	150 (100%)	
Female	0	0	
Smoking n (%)			<0.001 *
Yes	63 (52.5)	125 (83.3)	
No	57 (47.5)	25 (16.7)	
Dinking n (%)			0.934
Yes	69 (57.5)	87 (58.0)	
No	51 (42.5)	63 (42.0)	
Chronic disease n (%)			<0.001 *
Yes	51 (42.5)	106 (70.7)	
No	69 (57.5)	44 (29.3)	
Pneumoconiosis stage n (%)			
1		94 (62.7)	
2		47 (31.3)	
3		9 (6.0)	
Working age (years)		24.70 ± 8.48	
Complication n (%)			
Tuberculosis		19 (12.7)	
COPD		32 (21.3)	
Chronic bronchitis		56 (37.3)	
Two complications		4 (2.7)	
No complication		39 (26.0)	

* *p* < 0.05, significant difference between the two groups.

## Data Availability

The data presented in this study are available on request from the corresponding author. The data are not publicly available due to the regulations of the authors’ affiliations.

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
