# Peer review of "Screening of Serum Biomarkers of Coal Workers’ Pneumoconiosis by Metabolomics Combined with Machine Learning Strategy"

_ijerph, 2022, doi:10.3390/ijerph19127051_

Round 1

Reviewer 1 Report

The study investigates the possible role of metabolomics technology combined with machine learning strategy addressing the serum biomarkers of patients suffering from coal worker's pneumoconiosis (CWP). A case-control design was used and findings showed that the serum propylparaben could be a good biomarker able to identify the presence of the disease and correlate to its severity. The study is well conducted and well explained for reproducibility. In my opinion, the following points could be improved.

  1. Considering that propylparaben is still used in cosmetics, the possible role of gender should have been discussed: did the enrolled subjects screened for cosmetics use? Please add the gender distribution of the two groups.
  2. You considered some confounding factors including age, chronic diseases, smoking and drinking as matching variables between cases and controls (lines 79-80); you cited also "other factors" (line 80): could you better specify which factors you refer to? Moreover, environmental factors including the geographical locations where subjects spend mostly of their life time should be considered. Could you further specify this information? 
  3. Could you provide an hypothesis of a possible cut-off of propylparaben for the screening analysis of the CWP?
  4. In the Discussion section information on propylparaben should be shorter (lines 321-362).
  5. I suggest to implement a final paragraph of the role played by the new findings on possible implications in Public Health policies.

Minor editing corrections should be necessary:

  1. Please specify that the stages at lines 77-78 is referred to the disease;
  2. Please correct “elbow” vein (specifying which veins or in my opinion you can avoid citing it);
  3. Please check all units of measurement (e.g., “mM” at line 107: did you intend mMol?).

Reviewer 2 Report

Dear Authors,

The article makes an interesting contribution to the field on potential biomarkers of coal workers’ pneumoconiosis. This is a generally well written and well-structured manuscript. However, I have comments to the authors before the manuscript could be considered for publication.  

1.      Introduction should be revised and corrected so the reader would be clearly informed about the rationale of the study: What was the background and motivation to conduct the study? Could you, please, add what research on chemical substances related to coal worker's pneumoconiosis has been conducted so far in China.

2.      In the "Methods" section, there is no information when the study was performed - please indicate the period and the dates. More information about the hospital would also be appreciated, i.e. more data would be expected about the clinic (s), in which the data were collected.

3.      The first paragraph of the Discussion (lines 295-320) is far too long and should generally be a section with a few sentences describing the 'main findings'. I suggest that the first sentence strongly and clearly underlines the results achieved, especially on propylparaben. You can partially use the text from the lines 249-255, move it to the discussion and expand this text. Moreover, in the first paragraph of the Discussion there is a text on the strength of the study, which should be moved to the paragraph "Limitation and strength".

4.      In Discussion, it is also important to compare your study with the studies of other authors on potential pneumoconiosis’s biomarkers. Are there any other studies supporting your main result?

5.      Please, highlight the changes to the revised version using a different color and/or a different script.

Reviewer 3 Report

The manuscript reports the results of the research aiming to identify the omics biomarker of Coal workers' pneumoconiosis (CWP). The presented study found that the prototype propylparaben showed different levels in the serum of CWP patients at different stages, which may be related to changes in its metabolism. Propylparaben in cosmetics is the most important source of human exposure, as there are thousands of types of cosmetics using parabens as preservatives.

Specific remarks: 

  1. As the specific mechanism of propylparaben is unclear, the relationship between this chemical and CWP might be accidental. It would important for the reader to have more information about the possible relationship between propylparaben metabolisms and CWP development.
  2. The study was performed according to a case-control model and included 150 coal workers' pneumoconiosis (CWP) patients with different stages from two representative occupational disease hospitals in Beijing, China, and 120 healthy controls. We are informed that the subjects in the control group matched the case group with age, smoking, drinking, and other factors as much as possible (lines 79-80). On other hand, the Authors performed adjustments: including age, smoking, drinking, and chronic diseases (165-166). Please explain why the matching efforts were not effective, and statistical adjustments were needed
  3. In the introduction the authors report that “A coal still plays a dominant part in global energy 34 production and consumption, there is a very large number of people exposed to coal dust”(line 35). On other hand. The Authors explain that generally takes 10-20 years to develop pneumoconiosis after exposure to dust. It would be more appropriate to focus on the situation in the coal industry in that period, rather than on the current one.  
  4. Three machine learning (ML) methods were used in data analysis (RF, SVM, and Boruta). The results are displayed in several figures. Howler the flow f information is difficult to follow, especially for a reader not familiar with ML teachings.

Round 2

Reviewer 1 Report

Dear Authors,

Thank you for your considerable efforts. The main criticality is related to the difficulty in matching cases and controls, which is substantially absent. Your methodology of research is very interesting, but the choice of participants in the control group is uncorrelated to cases, and therefore the comparison is not effective. As you wrote among limitations (lines 429-438), you tried to match cases and controls without success, as confirmed by Table 1 (statistically significant p value for age, smoking, and chronic disease prevalence; and more, not considering work types and working age for the control group). Finally, in lines 149-151 you specified that the machine learning approach could not solve the matching problem ("grouping information as dependent variables"). If you are able to normalize your data using machine learning strategy as for statistical analysis, your findings will be valuable. Otherwise, I suggest you to repeat your analysis reviewing the participant selection process.

Moreover, the Discussion Section is still too long. For example, lines 309-315 could be removed as they are reported in the Methods section. I suggest you to sum up all propylparaben properties in two pages.

Finally, please change mM to mmol/L (line 106), which is the unit used in the International System of units (SI) (for details see: The Mole | LNE, Laboratoire national de métrologie et d'essais).
